# Study Design for an Evaluation of Newborn Screening for SCID in the UK

**DOI:** 10.3390/ijns8010004

**Published:** 2022-01-10

**Authors:** David Elliman

**Affiliations:** NHS England and Improvement and Great Ormond Street Hospital for Children, London WC1N 3JH, UK; David.elliman@nhs.net

**Keywords:** screening, TRECs, SCID, newborn, immunodeficiency, bloodspot

## Abstract

Severe combined immunodeficiency is a rare inherited disorder, which, if untreated, invariably proves fatal in late infancy or early childhood. With treatment, the prognosis is much improved. Early treatment of the siblings of cases, before they become symptomatic, has shown considerable improvements in outcomes. Based on this and the development of a test that can be used on the whole population of neonates (measurement of T-cell receptor excision circles—TRECs), many countries have added it to their routine newborn bloodspot screening programmes. The UK National Screening Committee (UKNSC) has considered whether SCID should be added to the UK screening programme and concluded that it was likely to be cost effective, but that there were a number of uncertainties that should be resolved before a national roll-out could be recommended. These include some aspects of the test, such as: cost; the use of different assays and cut-off levels to reduce false positive rates, while maintaining sensitivity; the overall benefits of screening for disease outcome in patients with SCID and other identified disorders; the need for a separate pathway for premature babies; the acceptability of the screening programme to parents of babies who have normal and abnormal (both true and false positive) screening results. To achieve this, screening of two thirds of babies born in England over a two-year period has been planned, beginning in September 2021. The outcomes and costs of care of babies identified by the screening will be compared with those of babies identified with SCID in the rest of the UK. The effect of the screening programme on parents will form part of a separate research project.

## 1. Introduction

This paper is based on a presentation made at a virtual meeting, entitled “Newborn Screening for SCID, ‘State of the Art’”, hosted by the International Society for Neonatal Screening and UK Newborn Screening Laboratories Network, January 2021. It has been updated in the light of progress since then, but prior to the launch of the programme in September 2021.

SCID is an inherited disorder of the immune system that renders the patient susceptible to what would usually be minor infections, which may prove fatal in patients with SCID. Immune reconstitution with haematopoietic stem cell transplantation (HSCT) or gene therapy have markedly improved the prognosis. A population screening test is now available that allows for the identification, pre-symptomatically, of patients with no family history [1]. Newborn screening for SCID was first implemented in the USA in 2008 [2]. Since then, newborn screening programmes have been rolled out across the whole of the USA and numerous other countries, including many European countries, New Zealand, Israel, and Taiwan [3]. In addition, other countries carry out screening only in part of the country or are conducting pilot studies. No case of SCID has been reported as missed by screening and the earlier treatment has improved prognosis. However, the programmes have produced a significant number of false positives, as well as identifying other immunological disorders, including idiopathic T-cell lymphopenia, where the benefits of screening are unclear.

In the UK, screening policy is decided in each nation by the respective Ministers of Health, based on the recommendations of the UKNSC [4]. The committee consists of a mix of professionals from different backgrounds and some lay members. Suggestions for screening are taken from any source and the details of consideration will depend on the strength of evidence from an initial review of the topic. A promising topic will be considered against a set of criteria [5], based on those drawn up for the WHO in 1968 by Wilson and Junger [6].

It is estimated that each year, of approximately 781,000 births in the UK, there would be 17 babies with SCID, i.e., about 1 in every 45,000 live births [7]. In 2017, the UKNSC received a commissioned report on the cost effectiveness of newborn screening for SCID. The report concluded that, overall, it appeared that newborn screening would be cost effective, but there were some uncertainties that could impinge on this conclusion [7] (pp. 15–17), [8] (pp. 14–15), namely the following:the cost of the TREC assay;the incidence of SCID;post-HSCT mortality rates in the early diagnosed population;the length of stay in hospital of the early diagnosed SCID patients;the proportion detected by family history in the absence of screening.

In addition, the committee felt some other issues should be explored, as follows:alternative assays for TRECs;number of babies who have low TRECs, but normal flow cytometry (false positives);the effect on the families of these babies;the desirability, or not, of a separate pathway for premature babies;the number of babies who have an abnormal assessment on flow cytometry and the categories of conditions detected;the extent of the benefit for babies with SCID;the extent of the benefit, if any, for babies with other conditions, particularly idiopathic T-cell lymphopenia.

The cost effectiveness report has been published [9]. As the ethnic mix in the UK is different from that of most countries where SCID screening currently takes place, and the health service is organized very differently, the committee recommended that an evaluation should take place, as it was important to be reasonably certain that a screening programme would be appropriate in the UK setting. It is recognized that some of the data will involve only small numbers; however, taken together with the international experience and some retrospective data, it should be adequate to allow a decision to be made.

## 2. Materials and Methods

To plan the evaluation and achieve these goals, a number of working parties were set up to design and monitor it. These included a laboratory group, a patient and professional information group, a data monitoring group, a clinical pathway group, and a diagnostic review group. Membership of each of these included all relevant stakeholders, including, for some, lay members. All reported to an overarching board.

Tasks included the following: procurement of equipment; drawing up of laboratory, referral, and clinical guidelines; training of staff; design of information materials for patients and professionals; running of webinars for staff (laboratory and clinical). In addition, the patient/family experience would be formally assessed in a research project.

In the UK, there are 13 newborn screening laboratories and a further 1 in each of the devolved nations. Based primarily on size, but also ensuring a broad ethnic representation, it was agreed that 6 English laboratories would screen for SCID during the evaluation. This amounts to about 400,000 babies per annum. Data would be collected for 2 years, amounting to approximately 800,000 babies. Details of children identified as having SCID in the non-SCID screening areas would also be included. The results would be analysed and a final decision as to whether the programme should continue and be rolled out over the whole of the UK would be made in the following 12–18 months. Largely due to the COVID-19 pandemic, the start of the evaluation was delayed until 6 September 2021. All babies whose blood spot cards are sent to participating laboratories from this date onwards will be included in the evaluation.

### 2.1. Pre-Evaluation Laboratory Work

At the start of the planning period, there was only one SCID screening method commercially available—the Perkin Elmer EnLite ^TM^ TREC assay. However, another became available soon after—the Immuno IVD SPOT-it ^TM^ TREC screening kit. It was decided to make use of the opportunity to evaluate both systems, with each being used in three of the six laboratories.

Initially, it was agreed that the procedure would reflect that which is normally followed internationally. Newborn samples, routinely taken on day 5 in England, would be assayed for TRECs. If the result exceeded a predetermined level (the analytical cut-off—“A” in Table 1), no further action would be taken and a result of “SCID not suspected” would be generated. Samples with low levels of TRECs would be re-tested in duplicate along with measurement of β-Actin. If the lower of these two TREC assays was below a second level (clinical cut-off—“B”), a result of “SCID suspected” would be produced. Any term baby with a TREC level lower than this and a satisfactory β-Actin level would be referred for an urgent clinical immunology assessment. If the β-Actin was unacceptable, a repeat sample would be requested.

Premature babies (<37 weeks gestation) with low levels of TRECs would have a repeat sample taken at 37 weeks gestation and then classified in the same way as a term baby. The management of premature babies changed during the pre-evaluation period (see a later section, including Figure 3) and the finally agreed pathway is that set out in Figure 1 and Figure 2.

The local expert immunology team will be phoned by the screening laboratory as soon as a low TREC result is available. The immunology team will arrange to see the baby and parents within the next 24 h. To reduce the period of uncertainty and anxiety for the family, the team would not contact parents on a Friday, at a weekend, or the day before a national holiday, to minimize the time between the parents being told the result and being seen. When seen, an overall assessment of the baby will be made and, at a minimum, blood would be taken for flow cytometry. Standard cut-offs are applied (see Figure 2).

TREC assays were carried out on a large sample of anonymized residual blood spots. This allowed population norms and cut-offs to be set for both assays to ensure that no cases of SCID were missed, yet the test was as specific as possible (see Table 1). 

The agreed cut-offs for flow cytometry were defined as ≥1500 CD3/μL and/or naïve T-cells ≥ 70%.

### 2.2. Prematurity

Initially, it was agreed that all babies with low levels of TRECs and normal β-actin levels would either be referred for an urgent diagnostic workup or discharged. This followed discussion where it was felt that this would allow a diagnosis to be reached as soon as possible and appropriate treatment to be instituted in a timely fashion, before infection had occurred. In addition, it would reduce the period of uncertainty for parents, whatever the ultimate diagnostic outcome. 

Prior to the start of the evaluation, 5000 anonymised samples were run to establish reasonable cut-offs that would avoid too many false positives. Values were determined for both methodologies. This pre-evaluation work showed that, in line with international experience, not treating premature babies separately would result in large numbers of false positive results overwhelming immunology services. Therefore, it was agreed that premature babies would have a different pathway (see Figure 3). Those with very low results (less than TREC level C) would be referred for a diagnostic work-up immediately, while those with a clearly normal result would need no further action. Babies with intermediate results would have their TRECs measured again at 37 weeks gestation or just prior to discharge from hospital, whichever came sooner.

### 2.3. Clinical Pathways for Those Who Have Abnormal Flow Cytometry

An accurate screening test without a high quality, timely, and nationally available clinical care pathway is of little value. The UK has the advantage of a nationally organized health service, free at point of contact, which facilitates uniform guidance and healthcare throughout the country. Management of babies with positive screening results will follow the same nationally agreed pathway, with few local variations [10].

### 2.4. SCID Screening and Vaccination

BCG is a live bacterial vaccine given to protect against TB. In the UK, BCG is offered selectively. The main group targeted are babies who reside in areas where the incidence of TB is equal to or exceeds 40 cases per 100,000 general population, or where a parent or grandparent was born in a high incidence country [11]. It is usually given in the maternity unit prior to discharge, i.e., before newborn screening has been undertaken. If a baby with a severe immunodeficiency, such as SCID, is given BCG, the organism may spread throughout the body, causing significant morbidity and it can sometimes be fatal [12]. If BCG were given to a baby who, subsequently on screening, was revealed to have SCID, specific treatment could be instituted and the adverse effects reduced, but not eliminated.

Before screening became widespread, consideration was given to delaying BCG vaccination until screening results were available [13]. The largest experience of newborn screening for SCID to date, has been in the USA, where BCG vaccination is not given. Practice in countries where screening for SCID has been introduced varies. Sweden offers the BCG vaccine selectively, normally after 6 months of age [14]. Taiwan introduced SCID screening in 2012, at which point the target age of the BCG vaccination universal policy was changed from 24 h of age to 1–4 weeks [15]. It has subsequently been changed to 5–8 months. Norway, one of the first countries in Europe to introduce newborn screening for SCID, recommends selective BCG vaccination at six-weeks old [16]. On the other hand, both Australia and New Zealand have newborn screening programmes that include SCID. Both recommend BCG that is given to selected populations in the neonatal period with no mention of waiting for screening results [17,18].

In the UK, the Joint Committee on Vaccination and Immunisation advise on vaccination policy. They advised that BCG vaccination should be postponed until the results of screening were available [19]. The target has been set for babies to be vaccinated when the screening results are available, which should be at or before 28 days of age. Arrangements have been put in place to ensure that BCG providers have access to the screening results.

Similar considerations apply to the rotavirus vaccine, which is given at eight and twelve weeks of age in general practice; however, the screening results should be available long before this.

The necessity to co-ordinate the introduction of the SCID evaluation with the change in the selective national vaccination programme has added a tier of complexity not encountered with other screening programmes.

### 2.5. The Patient/Family Experience

Patient information was designed by a group including public and professionals. It was then assessed by two focus groups of members of the public, before being amended. All pregnant women are given access, either hard copy or online, to this specific information about SCID, before screening takes place. This includes information about the condition, the screening test and its limitations, and the potential outcomes. It is explained that this is not part of routine care, but an evaluation, and they can choose to opt out of this without it affecting routine screening [20].

A research group has been commissioned to examine the patient experience. Carers of children who had true negative, false positive, and true positive screening outcomes will be included. Parents of babies ascertained via a family history or clinical presentation in non-screening areas, would also be included. In addition, a small group of carers of children with other disorders, revealed by screening, would also be interviewed. A potential outcome of screening is to find babies with idiopathic T-cell lymphopenia [21]. This has an indeterminate outcome, which could result in anxiety for carers. The carers of this group of children would be followed up along with some carers of children with an analogous designation from cystic fibrosis—CFSPID (cystic fibrosis screen positive indeterminate status) [22]. All families in the research study would be followed up until the children were five years old. Only the early data would feed into the evaluation, but later data might help remodel the programme and be useful in the formulation of other screening programmes.

### 2.6. Information for Professionals

The same multidisciplinary group that produced materials for pregnant women also produced materials for professionals likely to be involved in administering or advising on the programme. These materials include, inter alia, blogs, webinars, slide sets, and letters updating colleagues on progress and policies [23].

### 2.7. Data Collection

#### 2.7.1. Clinical Data

Screening laboratories will report aggregate date on all babies screened with a normal TREC result, along with individual data on all screened positive babies. These data will be stored on a secure national database with limited access. Data on babies ascertained not via screening, will also be collected throughout the UK. Final diagnoses/designations will be scrutinized by a diagnostic review group.

Data will be examined at regular intervals to ensure that no alterations are needed to the programme to ensure that it is on target to attain its goals of an accurate and timely outcome for patients and records data sufficient for the needs of UKNSC.

#### 2.7.2. Resource Usage

To populate the cost effectiveness analysis required by UKNSC, information will be collected on the resources and their costs used in screening, diagnosis, and treatment.

## 3. Conclusions

Planning the evaluation has proved to be extremely complex. The technology used is new to screening and the interdependencies with many groups of professionals, as well as the public, have meant the planning of the evaluation itself has been resource intensive. Without the input of these groups and the careful planning, it would not be possible to organize an evaluation ensuring a high quality, timely service to patients, meeting the data needs of the UKNSC.

## Figures and Tables

**Figure 1 IJNS-08-00004-f001:**
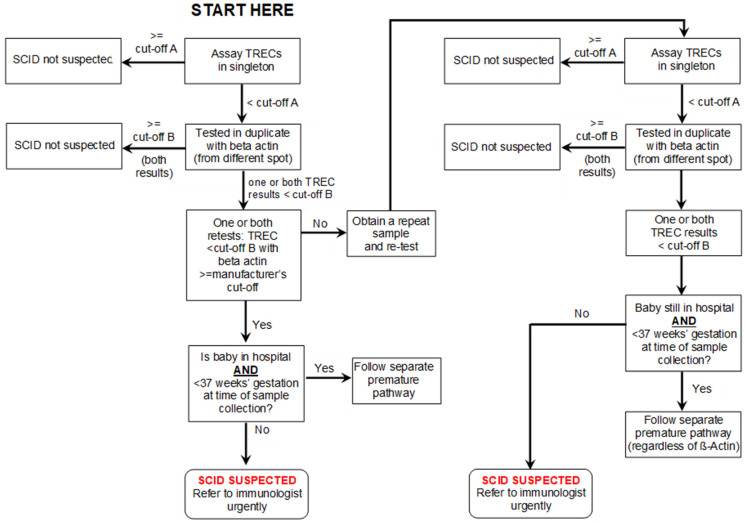
SCID screening algorithm (see Table 1 for cut-off values).

**Figure 2 IJNS-08-00004-f002:**
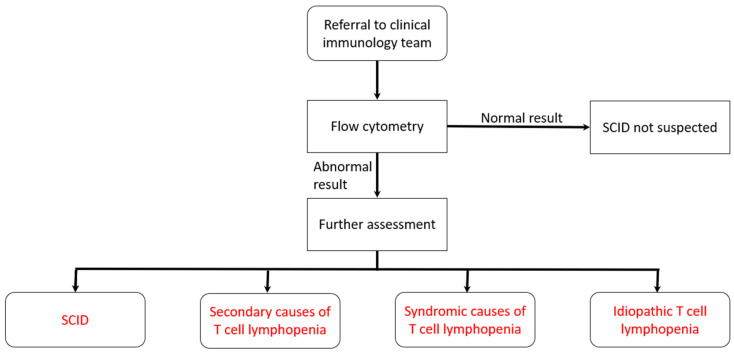
Pathway for babies referred to clinical immunology team (see box for cut-offs).

**Figure 3 IJNS-08-00004-f003:**
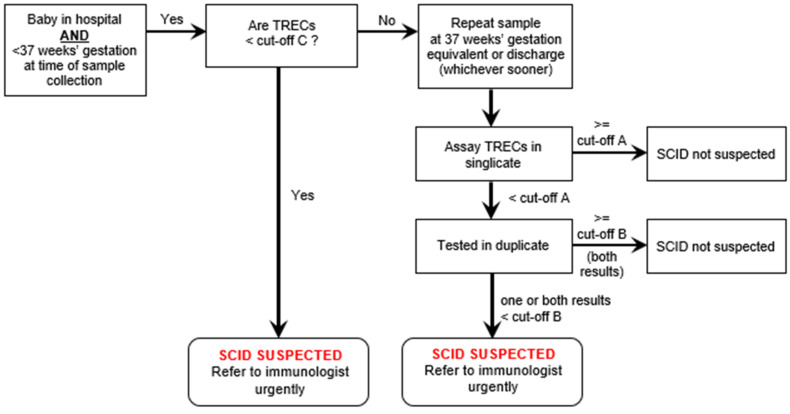
Pathway for premature babies found to have low TRECs on first assay.

**Table 1 IJNS-08-00004-t001:** Cut-off values for TRECs and β-Actin.

		Perkin Elmer EnLite ^TM^ (Copies/μL)	Immuno IVD SPOT-It ^TM^ (Copies/Punch)
TRECs	A	30	12.0
	B	20	8.0
	C *	8	4.0
β-Actin		53	1000

* TREC level C applies to premature babies only (see below).

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
