# Peer review of "Study Design for an Evaluation of Newborn Screening for SCID in the UK"

_2409-515X, 2022, doi:10.3390/ijns8010004_

Round 1

Reviewer 1 Report

The manuscript gives a clear description of the study design for the NBS for SCID in the UK.

I only have some minor comments:

  1. What is the expected incidence of SCID cases in the regions were NBS SCID is implemented? Please add this information somewhere in the document. Furthermore, will this number be high enough to solve the uncertainties (listed on page 2) related to early diagnosed SCID patients?
  2. Introduction:  which presentation in January from which year is meant?
  3. Introduction: can you add references of countries that have introduced newborn screening with good results? What is meant by good results?
  4. Introduction: ref 5, pp 15-17 instead of 15-170.
  5. Introduction: Is there also a reference to the report on the cost-effectiveness of the Committee (which committee?) itself? Now only references to meeting notes [5,6] are given.
  6. Material and methods: what is meant by the start of the evaluation (6th September 2021), is this the start of data collection?
  7. Material and methods, Box: What is meant by the cut-off values of  TRECs C? Only cut-off A and B are mentioned before.
  8. Material and methods, 2.4: The target has been set that babies should be vaccinated(?) when the screening results are available....

Author Response

I am grateful for the reviewer's comments which will improve the quality of the manuscript.

  1. What is the expected incidence of SCID cases in the regions were NBS SCID is implemented?  Please add this information somewhere in the document. I have added a sentence. Furthermore, will this number be high enough to solve the uncertainties (listed on page 2) related to early diagnosed SCID patients? It is a moot point. I have added a sentence, which I hope addresses this.
  2. Introduction:  which presentation in January from which year is meant? I have clarified this.
  3. Introduction: can you add references of countries that have introduced newborn screening with good results? What is meant by good results? I have included a brief background section which I hope answers this question.
  4. Introduction: ref 5, pp 15-17 instead of 15-170. Corrected
  5. Introduction: Is there also a reference to the report on the cost-effectiveness of the Committee (which committee?) itself? Now only references to meeting notes [5,6] are given. The cost effectiveness report went to the UKNSC. I have now referenced the published version of the report.
  6. Material and methods: what is meant by the start of the evaluation (6th September 2021), is this the start of data collection? I have now clarified this.
  7. Material and methods, Box: What is meant by the cut-off values of  TRECs C? Only cut-off A and B are mentioned before. I have added a note to the box and also to section 2.2
  8. Material and methods, 2.4: The target has been set that babies should be vaccinated(?) when the screening results are available.... Corrected

Reviewer 2 Report

One comment which may improve the manuscript:

In section 2.4 could the international experience with SCID NBS and BCG/rotavirus vaccination strategy be given for context?

Two typos:

Section 2.1 "1,500 CD3/μ and/or naive cells" should be "1,500 CD3/μand/or naive T cells".

Section 2.1 

Author Response

Thank you. I have corrected the typos.